# GNN Codon Adjacency Tunes Protein Translation

**DOI:** 10.3390/ijms25115914

**Published:** 2024-05-29

**Authors:** Joyce Sun, Pete Hwang, Eric D. Sakkas, Yancheng Zhou, Luis Perez, Ishani Dave, Jack B. Kwon, Audrey E. McMahon, Mia Wichman, Mitsu Raval, Kristen Scopino, Daniel Krizanc, Kelly M. Thayer, Michael P. Weir

**Affiliations:** 1Department of Biology, Wesleyan University, Middletown, CT 06459, USA; jsun02@wesleyan.edu (J.S.); phwang@wesleyan.edu (P.H.); esakkas@wesleyan.edu (E.D.S.); ezhou@wesleyan.edu (Y.Z.); lperez@wesleyan.edu (L.P.); idave@wesleyan.edu (I.D.); jkwon@sail.bio (J.B.K.); aemcmahon@wesleyan.edu (A.E.M.); mwichman@wesleyan.edu (M.W.); mraval@wesleyan.edu (M.R.);; 2Department of Mathematics and Computer Science, Wesleyan University, Middletown, CT 06459, USA; dkrizanc@wesleyan.edu (D.K.); kthayer@wesleyan.edu (K.M.T.); 3College of Integrative Sciences, Wesleyan University, Middletown, CT 06459, USA; 4Department of Chemistry, Wesleyan University, Middletown, CT 06459, USA

**Keywords:** protein translation, codon adjacency, wobble base pairing, ribosome CAR surface, ribosome translocation

## Abstract

The central dogma treats the ribosome as a molecular machine that reads one mRNA codon at a time as it adds each amino acid to its growing peptide chain. However, this and previous studies suggest that ribosomes actually perceive pairs of adjacent codons as they take three-nucleotide steps along the mRNA. We examined GNN codons, which we find are surprisingly overrepresented in eukaryote protein-coding open reading frames (ORFs), especially immediately after NNU codons. Ribosome profiling experiments in yeast revealed that ribosomes with NNU at their aminoacyl (A) site have particularly elevated densities when NNU is immediately followed (3′) by a GNN codon, indicating slower mRNA threading of the NNU codon from the ribosome’s A to peptidyl (P) sites. Moreover, if the assessment was limited to ribosomes that have only recently arrived at the next codon, by examining 21-nucleotide ribosome footprints (21-nt RFPs), elevated densities were observed for multiple codon classes when followed by GNN. This striking translation slowdown at adjacent 5′-NNN GNN codon pairs is likely mediated, in part, by the ribosome’s CAR surface, which acts as an extension of the A-site tRNA anticodon during ribosome translocation and interacts through hydrogen bonding and pi stacking with the GNN codon. The functional consequences of 5′-NNN GNN codon adjacency are expected to influence the evolution of protein coding sequences.

## 1. Introduction

Elevated frequencies of GCN codons (where N = A, C, G or U) have been noted in prokaryote and eukaryote ORFs [1,2,3,4,5] (Figure 1), and their complementarity to sequences in the 530 loop of the ribosome’s 16S rRNA has led to speculation that this nucleotide periodicity might be associated with the regulation of protein translation [2,3,4,5]. With publication of cryo-EM structures of translocating yeast ribosomes, Abeyrathne et al. [6] noted hydrogen bonding between 18S rRNA C1274 and the mRNA nucleotide following (3′ of) the A-site codon. With further examination, we discovered that C1274 is part of CAR [1,7,8,9], a three-residue surface that interacts with the full (+1) codon 3′ adjacent to the A-site codon. CAR consists of yeast 18S rRNA C1274 and A1427—which are conserved in prokaryotes and eukaryotes and are henceforth referred to as C1054 and A1196, the corresponding 16S rRNA residues in *E. coli*—and R146 of the yeast ribosomal protein Rps3, which is only conserved in eukaryotes. The residues of CAR pi stack with each other; C1054 of CAR also pi (base) stacks with nucleotide (nt) 34 of the A-site tRNA anticodon, the partner of the A-site wobble base. This stacking of nt 34 with C1054 of CAR positions CAR as an extension of the anticodon and in a location where it can interact with the mRNA +1 codon. Molecular dynamics (MD) simulations of a subsystem of the decoding center neighborhood of translocating ribosomes revealed that the interaction between CAR and the +1 codon is particularly strong during stage II of translocation [9], and the interaction showed a remarkable sequence preference for GNN codons [7,8,9]. The discovery of CAR and its sequence-specificity for +1 GNN codons further supported the hypothesis that GNN codons might participate in a layer of protein translation regulation. Through a context-sensitive analysis of published ribosome profiling data, we show here that this is indeed the case. Because CAR is anchored through base stacking to tRNA nt 34, we classified codons/anticodons according to their nt 34 and wobble nucleotide identities and found that NNU A-site codons showed particularly pronounced slowdown in ribosome translocation when immediately followed by a +1 GNN codon.

## 2. Results and Discussion

### 2.1. NNU Codons Have Distinctive Properties

Each NNU codon shares the same tRNA with its corresponding NNC codon containing the same nucleotide identities at positions 1 and 2 of the codon. Since these NNU/NNC codon pairs experience the same abundances of their shared tRNAs, this raises the question of whether their translation properties are similar. The anticodons of these shared tRNAs have either a G34 or I34 (inosine 34) nucleotide, both of which can base pair with the wobble U or C nucleotides of the mRNA A-site codon. I34 (adenine 34 converted to inosine 34) is a eukaryote innovation that is used by NNU/C codon pairs in cases where the corresponding NNA codon also codes for the same amino acid (because inosine can also base pair with wobble A [11]). Interestingly, the CGA codon, which has particularly slow translation [12,13], is the only yeast NNA codon that relies exclusively on an I34 anticodon.

NNU codons are significantly more abundant than NNC in yeast ORFs (Appendix A), although for NNU codons that utilize tRNA G34, this trend is less strong in ORFs of genes with high protein expression (Appendix A). Unlike other codon classes, NNU codons are significantly more likely to be immediately followed by a GNN codon (Figure 1 and Appendix A), and this trend is particularly strong for ORFs of genes with high protein expression (Figure 1, Top 10%), suggesting that this codon adjacency may have functional significance. Indeed, position weight matrices for coding sequences of multiple eukaryotes revealed elevated GNN frequencies after NNU codons (Appendix A), whereas testing of several prokaryote coding sequence showed elevated GNN frequencies after both NNU and NNC codons (Appendix A). While these GNN codon frequency and adjacency properties could reflect multiple selection pressures, including the evolution of preferred amino acid identities and adjacencies in protein sequences, here, we explore the possibility that protein translation mechanisms could be sensitive to GNN codon sequences which are implicated in the function of the CAR ribosome surface [7,8,9], as discussed below.

We note that yeast and other species also show pronounced underrepresentation of NGN codons (Figure 1 and Appendix A). This G2 depression is mainly present in UNN and CNN codon classes (Appendix A), possibly reflecting, in part, selection against stop codons in ORFs, as well as observations in yeast [12,13,14] that CGN codons are translated slowly and adjacent pairs of codons that inhibit translation often contain a CGA codon. The inhibitory codon pairs stall in the ribosome’s A and P sites and the accommodation of the A-site tRNA appears disrupted by mis-positioning of the first nucleotide of the A-site codon and a 3′ shift of one nucleotide in the placement of the mRNA kink between the P and A sites, which is normally important for reading frame maintenance [14].

### 2.2. GNN Adjacency Slows Protein Translation

The hints that NNU GNN codon adjacency might be related to ribosome function led us to examine the protein translation rates of NNU codons with and without adjacent GNN codons. Protein translation rates were assessed by looking at ribosome footprint (RFP) densities on mRNAs, which provides a proxy for translation rate where higher density implies slower translation [15] because codons with slower translation receive more footprint “snapshots”.

We analyzed published yeast ribosome profiling data [16] that utilized the translation inhibitors tigecycline (TIG) and cycloheximide during the harvesting of cell extracts to ensure the synchronous cessation of ribosome movement along the mRNA. We examined NNU and NNC codons located at the ribosome A site and compared the ribosome densities for NNU/NNC codon pairs where the codons were followed or not followed by +1 GNN. For all NNU codons (except AUU and CCU, see Appendix A), ribosome densities were significantly higher when the codon was adjacent to a 3′ +1 GNN codon compared to when it was not followed by a +1 GNN codon (bootstrap *p* < 0.01; blue bars in Figure 2 and replicate data set in Appendix A). These trends were observed whether the NNU codon utilized tRNA G34 or I34. Similar consistent trends were not observed for NNC codons (Figure 2 and Appendix A, orange bars; Appendix A) or with NNA A-site codons (except GUA codons, Appendix A). However, several A-site NNG codons had significantly elevated ribosome densities when followed by +1 GNN, and two (AGG and CAG) showed significantly depressed densities (see Appendix A legend, Figure 3 and Appendix A). The pronounced density increases for most NNU codons are consistent with previous observations [17] that codons with wobble U:G base pairing are translated slowly.

The above analysis was performed for ribosome footprints of 27–30 nucleotides (28-nt RFPs), which map ribosomes with tRNAs at their P and A sites. Ribosomes also exhibit shorter 20–22 nt footprints (21-nt RFPs), which represent pre-accommodation ribosomes that have not received an A-site tRNA and were frozen in this state because the cell lysates were treated with TIG, which blocks tRNA accommodation [16]. Although fewer genes had above-threshold densities of the shorter footprints (>1 footprint per 10 nt), our examination of 21-nt RFPs showed that many classes of codons at the A-site had higher ribosome densities when followed by +1 GNN (bootstrap *p* < 0.01; Figure 3, Appendix A). Most NNU codons showed high increases in density, including NNU codons that utilize I34 tRNAs (Figure 3, Appendix A). Also of note, many NNC and NNA codons, which did not show significant +1 GNN-mediated density increases for 28-nt RFPs, nevertheless had significant density increases for the 21-nt RFPs. These observations suggest that there is a general slowing of pre-accommodation ribosomes when A-site codons are followed by +1 GNN.

### 2.3. The CAR Surface Is Ideally Positioned to Mediate +1 GNN Regulation

The mechanisms by which pre-accommodation ribosomes (21-nt RFPs) are slowed by +1 GNN codons are unknown. However, once ribosomes have accommodated the correct A-site tRNA (28-nt RFPs), the effects of +1 GNN codons on translation rates are likely mediated by the ribosome CAR surface, which is located next to the A-site tRNA anticodon during translocation [1,7,8,9]. Supporting this hypothesis, atomic granularity MD analysis (see Section 3.2) of the translocation ribosome decoding center with A-site NNU codons followed by +1 GCU or +1 CGU revealed significantly higher H-bond interactions between CAR and +1 GCU compared to +1 CGU (Figure 4A,B; Student’s *t*-test *p* < 0.01). These effects were most pronounced for NNU codons that utilized tRNA G34 and were more muted for those that utilized tRNA I34. A-site NNC codons also showed higher H-bonding between CAR and +1 GCU than +1 CGU, although these effects were less pronounced than for corresponding NNU codons (Figure 4A,B). Our analysis included assessment of the same A-site codon pair (CCU/CCC) tested with tRNA G34 or tRNA I34—the former is utilized by bacterial and archaeal tRNAs and the latter by eukaryotes—and this direct comparison of otherwise identical ribosome subsystems confirmed that I34 led to more muted differences between +1 GCU and +1 CGU compared to G34 (Figure 4A,B).

Our MD analysis also revealed characteristic pi and pi–cation stacking behaviors of CAR (Figure 4D–F, Appendix A). G34 or I34 pi stacked with C1054 of CAR for both A-site NNU and NNC codons with either +1 GCU or +1 CGU. Moreover, I34, a eukaryote innovation, showed significantly more stacking than G34 (Appendix A; Student’s *t*-test *p* < 0.05). We also observed frequent stacking between nucleotide 2 of the +1 codon and either A1196 or R146 of CAR: the cytosine of +1 GCU tended to pi stack with A1196 of CAR (Figure 4E), and the guanine of +1 CGU often cation–pi stacked with R146 of CAR (Figure 4F).

### 2.4. A Sequence-Sensitive Ribosome Braking System

Each time the ribosome takes a translocation step, the A-site tRNA and its base-paired mRNA codon need to be released from the A site to move across to the ribosome P site. Since CAR is part of the ribosome structure that remains behind as the tRNA and mRNA advance (Figure 4C), any modulation of the strength of the interaction between CAR and the +1 codon could influence the speed of release. We speculate that the differences in H-bonding and pi stacking interactions associated with +1 GCU and +1 CGU could lead to differences in the rates of ratcheting movements of the ribosome during translocation. Even if subtle, the differences summed over multiple codon pairs could lead to substantial effects on overall protein translation rates. Clustering analysis of cryoEM images [6] has defined five stages of translocation of the yeast ribosome. The interaction between CAR and NNU +1 codons is strongest at translocation stage II and then gradually decreases as the ratcheting proceeds onwards through stages III to V as the A-site tRNA and its associated mRNA codon translocate across to the P site [9]. Stronger interactions of CAR with +1 GNN codons may act like a transient brake, slightly reducing the probability of CAR “letting go” of the mRNA +1 codon, thereby slightly reducing the rate of translocation (Figure 4C). With translation elongation rates in eukaryotes estimated to be in the order of five codons per second [18], since the ribosome footprint densities of NNU codons with +1GNN are on average about 40% higher than without (Figure 2, Appendix A), their average elongation rates are likely correspondingly slower.

The adjacency of several combinations of codons has previously been implicated in the regulation of protein translation [12,13,19], and abnormal structures with adjacent CGA and CCG (or CGA) at the ribosome’s P and A sites, respectively, have been associated with ribosome stalling [14]. Our new observation involves adjacent codons in the ribosome’s A and +1 sites and applies to the large majority of codons as they arrive at the decoding center A site, switching between slower and faster translation depending upon whether or not the +1 codon starts with a G nucleotide. For A-site NNU codons, this effect is particularly strong and extends into the translocation stages of elongation. The discovery of the ribosome’s CAR surface, which behaves as an extension of the A-site tRNA anticodon, provides a potential structural basis for ribosome slowdown. The striking sequence specificity of the CAR effect, both with respect to the wobble nucleotide/tRNA nt 34 identities as well as the sequences of the A-site and +1 codons, suggests that codon adjacencies may have evolved in ORFs to favor efficient protein translation. Selection for GNN-mediated translation slowdown could aid in tuning the cotranslational folding of the growing polypeptide chains [20,21,22] or in reducing the probabilities of ribosome–ribosome collisions and resulting stress responses [23] in mRNA regions prone to ribosome crowding. The potential selective advantages of codon adjacencies for cells in different contexts await future analysis. Indeed, the centrality of tRNA nt 34 in anchoring CAR to the anticodon is of particular interest since this residue can be differentially modified under different growth conditions, including stress [24,25,26,27], potentially affecting CAR function.

## 3. Materials and Methods

### 3.1. Ribosome Profiling Analysis

Our analysis used published ribosome profiling data [16] which had high-confidence mapping of ribosome A sites to mRNA sequences because, upon the harvesting of yeast cells, the progress of ribosomes along the mRNA was blocked effectively by the antibiotics tigecyclin (TIG) and cycloheximide (CHX). TIG blocked the recruitment of new tRNAs into the ribosome A site [28], and CHX blocked the translocation of A-site tRNAs to the P-site [29]. Ribosomes that have recruited an A-site tRNA give rise to 27–30 nt footprints (28-nt RFPs), whereas ribosomes that have not yet accommodated a correct tRNA at the A site give rise to shorter 20–22 nt footprints (21-nt RFPs) [16]. The combination of TIG and CHX effectively froze in time the positions of ribosomes on mRNAs, allowing good quantitative assessment of ribosomes positioned on each A-site codon. Ribosome footprint sequences were aligned with mRNA sequences using STAR 2.5.3a Aligner [30]. The alignments of 28-nt RFPs for higher-expression genes were improved with the Ribodeblur algorithm [31]. A-site codons were identified using offsets from the 5′ end of the footprints, as described for 21-nt RFPs [16] and 28-nt RFPs [31]. As discussed previously [15], ribosome profiles show three-nucleotide periodicity, and the densities at individual codons were computed by taking the major peaks in normalized ribosome counts and adding the normalized counts from adjacent minor peaks on either side—whose intensities were found to correlate strongly with the intensities of the major peaks. Ribosome counts were normalized to the average codon densities for each gene, and the codons in each gene were given equal weighting in our analysis to codons from all other genes. Using WT1 and WT2 replicate data sets [16], our analysis was limited to genes (ORFs > 198 nt) with average footprint densities above 1 footprint per 10 nt (WT1: 2132 genes, WT2: 1279 for 28-nt RFPs; WT1: 1591, WT2: 1100 for 21-nt RFPs). Statistical significance of differences in A-site codon densities with and without +1 GNN codons was assessed with bootstrap analysis using mean ribosome densities from 10,000 samples of 2132, 1279, 1591 or 1100 genes randomly chosen with replacement. The mean density for codons with +1 GNN was compared with the bootstrap distribution of mean densities for codons without +1 GNN. 

### 3.2. Molecular Dynamics Simulations

Atomic-level MD analysis allows simulation of the forces experienced over time by atoms in a molecular structure. Our MD experiments were performed as described in Dalgarno et al. [7]. Briefly, we used a 494-residue subsystem of the ribosome consisting of a decoding center neighborhood. Residues at the periphery of the subsystem were restrained to retain the conformation of stage II of yeast ribosome translocation (PDB ID 5JUP [6]). Nucleotide identities in the A-site codon and anticodon were changed using AMBER22 tLEaP, and 30 independent replicate trajectories (20 × 60 ns; 10 × 100 ns) were performed for each tested version of the subsystem. RMSD analysis confirmed that trajectory behavior had settled by 20 ns, and the trajectory frames after 20 ns were analyzed with cpptraj functions to measure H-bonding between the A-site and +1 codons and the anticodon and CAR surface, and to measure base stacking using distances between the centers of geometry of stacked base rings or the guanidinium group of R146 (of CAR). For purine bases, we used the shorter of the distances from the pyrimidine or imidazole ring. Statistical significance was determined with Student’s *t*-tests of the 30 MD replicates.

### 3.3. Codon Frequency and Information-Theoretic Analysis

The analysis of codon frequencies and the construction of position weight matrices for codons following reference codons was coded in Python and graphed with Excel and R. Nucleotide frequencies were used to compute position weight matrices (where weight = log_2_(f_observed_/f_expected_)). Codon frequency differences between NNU and NNC codons were tested by bootstrap analysis of gene samples with replacement. A similar bootstrap analysis was performed to assess the differences in weights for adjacent codons with or without G at position 1 of the +1 codon. Significance thresholds (two-tailed) in the bootstrap distribution of G1 weights for all codons were used to test whether codons following NNU or NNC had significantly different weights.

## Figures and Tables

**Figure 1 ijms-25-05914-f001:**
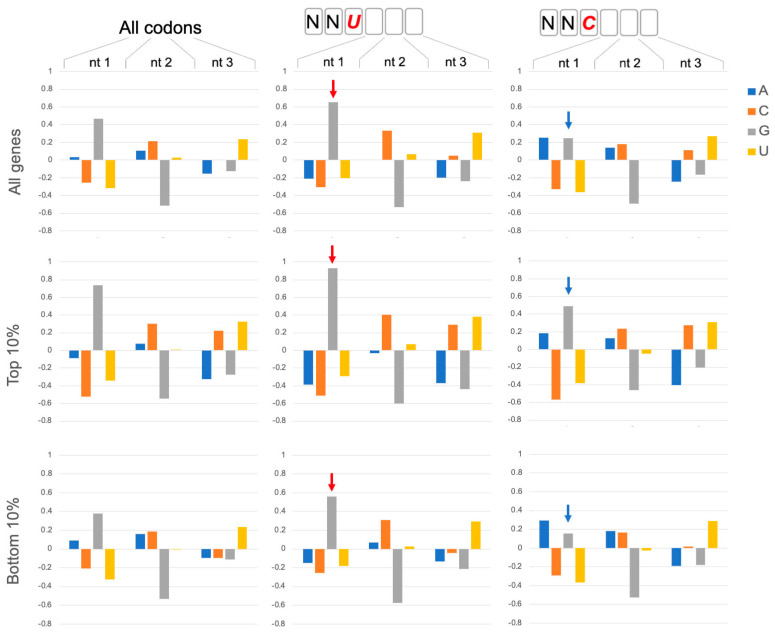
GNN codon adjacency. Codon nucleotide preferences are illustrated by log_2_(f_observed_/f_expected_) using background (expected) frequencies of nucleotides in ORFs regardless of codon position (A:0.326; C: 0.192; G: 0.204; U:0.278). Assessment of all codons in 6703 coding genes in *Saccharomyces cerevisiae* (All codons) showed that G is enhanced in the first nucleotide of codons. This G1 enhancement is significantly more pronounced in codons that are 3′ adjacent to NNU codons (red arrows; bootstrap *p* < 0.01) and less pronounced adjacent to NNC codons (blue arrows; bootstrap *p* < 0.01). The preference for GNN codons is even more pronounced (bootstrap *p* < 0.01) in genes with high protein expression: the top 10% of protein expressers (Top 10%; 388 genes) from a set of 3868 genes with detectable protein expression in a genomic-scale western and reporter gene analysis in yeast [10]. The bottom 10% of protein expressers (381 genes) have lower G1 enhancement but show the same trends with more (less) pronounced G1 enhancement after NNU (NNC) codons (bootstrap *p* < 0.01). The codon position weight matrices also reveal other striking trends, including strong disfavoring of G at position 2 of codons, as noted previously [1].

**Figure 2 ijms-25-05914-f002:**
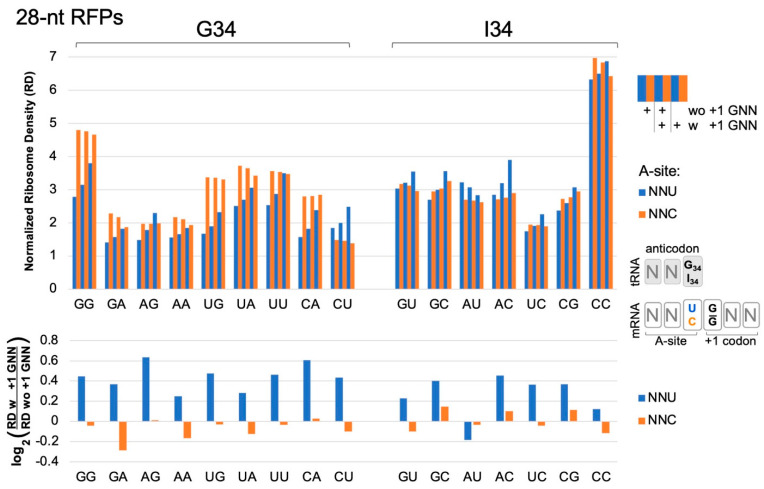
Ribosome densities at A-site NNU/C codons. Analysis of published [16] ribosome profiling data for 629,141 footprints of length 27–30 (28-nt RFPs) in ORFs of 2132 genes. A-site NNU codons (blue bars) that utilize tRNA G34 or I34 have higher ribosome densities (except AUU, see below), suggesting slower translation rates, when immediately followed (3′) by a codon with G at its first position (with (w) +1 GNN compared to without (wo) +1 GNN). NNC codons (orange) do not show these trends. Codons are labeled on the x-axis according to their first two nucleotides. Codon ribosome densities (RD) were normalized relative to other codons in the same mRNA. Similar results were observed in an independent replicate experiment from the same study [16] (WT2, Appendix A). Bootstrap analysis (2-tailed, *p* < 0.01) confirmed that in both replicate experiments, all NNU codons, except AUU and CCU, had higher densities when followed by +1 GNN (see Appendix A). CCU had higher densities in one replicate; AUU had lower densities in one replicate. No NNC codons had consistently higher or lower densities in both replicates.

**Figure 3 ijms-25-05914-f003:**
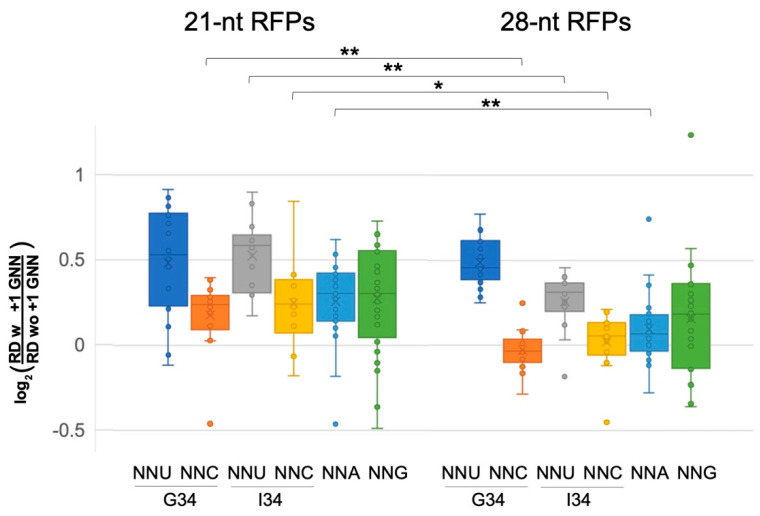
Pre-accommodation and translocation ribosomes are slowed by +1 GNN codons. Analysis of 21-nt ribosome footprints (RFPs) showed that +1 GNN codons led to elevated ribosome densities for all codon classes, and particularly for A-site NNU codons. The 28-nt RFPs showed similar trends, but +1 GNN codons led to negligible density increases for NNC and NNA codons (see Appendix A). The density distributions for 21- and 28-nt RFPs were compared using 2-tailed Student’s *t*-tests (*p* < 0.05 *; *p* < 0.01 **): compared to the 21-nt RFPs, +1 GNN codons were associated with lower increases in 28-nt RFP densities for A-site NNC, NNA and NNU (G34). The 21-nt RFPs likely represent pre-accommodation ribosomes that have not accommodated a tRNA at their A-site, whereas the 28-nt RFPs represent translocating ribosomes that have A- and P-site tRNAs. Data were collected from two independent replicate experiments [16] (WT1 and WT2, Figure 2, Appendix A), and the average normalized ribosome densities for each codon type (one for each replicate) were included in the whisker plot. However, CGA 21-nt RFPs for replicate WT2 were excluded due to very small numbers of CGA GNN codon pairs with above-threshold ribosome densities (>1 footprint per 10 nt).

**Figure 4 ijms-25-05914-f004:**
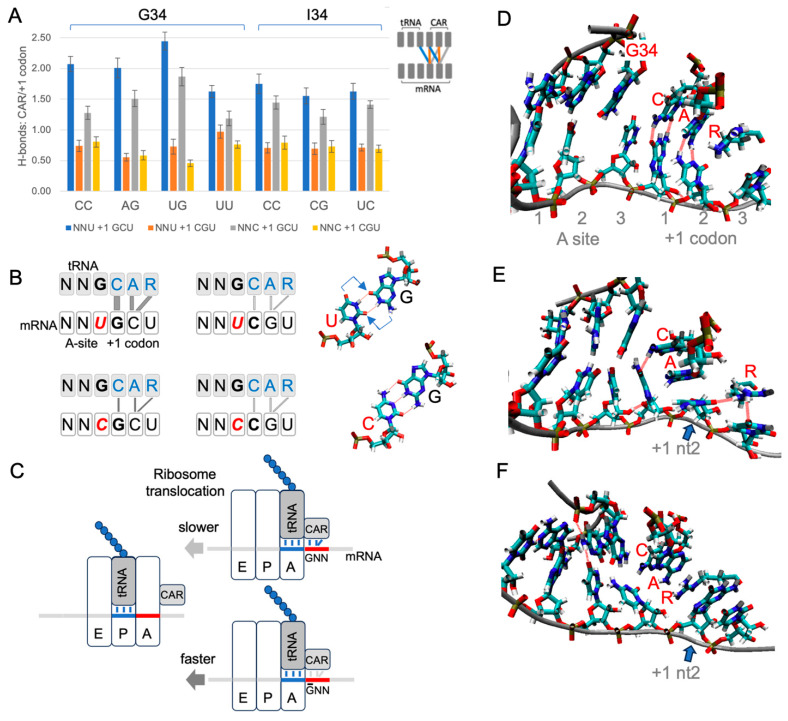
CAR is positioned to mediate +1 codon translation regulation. (**A**) H-bonding levels between CAR and the +1 codon (see schematic insert and panel **B**) were assessed in MD trajectories for different combinations of A-site and +1 codons in a subsystem of the translocation stage II ribosome (each construct was tested using 30 MD replicates). The CAR surface preferentially H-bonds in subsystems with +1 GCU codons, and H-bonding is more pronounced with A-site NNU codons that utilize tRNA G34. (**B**) Schematic summary showing stronger H-bonding between CAR and +1 GCU codons. Wobble U:G base pairing is displaced (arrows) compared to C:G, potentially influencing CAR behavior. (**C**) Stronger interactions between CAR and the +1 GNN codon are hypothesized to slow threading of the mRNA when the A-site tRNA and its base-paired codon translocate to the ribosome P site. (**D**) Frame of MD trajectory showing H-bonding between CAR and +1 codon nucleotides 1 and 2 (broken red lines; A-site UUU +1 GCU shown here). (**E**) In subsystems with +1 GCU, A1196 of CAR often stacks with +1 codon nucleotide 2 (blue arrow; A-site UUU +1 GCU). (**F**) Subsystems with +1 CGU frequently show stacking of R146 of CAR with +1 codon nucleotide 2 (blue arrow; UGU +1 CGU; also see Appendix A).

## Data Availability

Data and code are available at https://doi.org/10.25438/wes02.25328632.v1 (accessed on 24 May 2024). ORF analysis: code for sequence and information-theoretic analysis; ribosome profile analysis: +1 GNN codon analysis and data; MD experiments: restart structure and topology files and code for molecular dynamics experiments.

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
