# Peer review of "GNN Codon Adjacency Tunes Protein Translation"

_ijms, 2024, doi:10.3390/ijms25115914_

Round 1

Reviewer 1 Report (Previous Reviewer 2)

Comments and Suggestions for Authors

The authors have addressed the technical concerns raised during the review of the original version. In this reviewer's opinion, the revised version is appropriate for publication in the Molecular Informatics section of IJMS.

Author Response

not applicable

Reviewer 2 Report (Previous Reviewer 1)

Comments and Suggestions for Authors

The authors have responded to previous concern and the manuscript is now acceptable for publication.

That said, as a final note, the authors should consider their use of “translated slowly” when referring to the effect of GNN codons.  How much more slowly does translation occur when the GNN codon is encountered (i.e. as the normal transit time for each amino acid averages about 0.2 seconds for eukaryotes at 37o C, is slowly 0.3 seconds, 0.5 seconds, 1 second)?  The authors may be able to assess this change based upon their analysis of ribo-seq data.

Author Response

This manuscript is a resubmission of an earlier submission. The following is a list of the peer review reports and author responses from that submission.

Round 1

Reviewer 1 Report

Comments and Suggestions for Authors

This is an interesting manuscript that describes the probability of finding a GNN codon to the 3’ side of an A site.  Arguments are made that having a G in this position enhances interactions with the “CAR” surface of the 40S subunit.  However, this observation is not reviewed with respect to tRNA populations or positional placement in the mRNA (i.e. an intentional pause site to allow for folding of the growing polypeptide chain).  This reviewer feels that the authors need to consider several other elements in their analysis.

Concerns

1.        From the text book on Biochemistry (Voet and Voet, 1990), the average amino acid composition of amino acids that have a G as the first nucleotide in a codon are: Gly-7.5%, Ala – 9,0%, Val – 6.9%, Asp – 5.5% and Glu – 6.2% for a total of 35.1%).  Thus it would be anticipated that a GNN codon would be more likely to occur than random chance for 20 amino acids.

2.        There is also the consideration of opthamality (a reflection of tRNA abundance).  From Presnyak et al., 2015, Figure 2, five of the top six most optimal codons begin with G (GCT, GGT, GTC, GTT, and GCC).  Does this play a role in your observations?

3.        Page 3, line 96 – “translation rate where higher density implies slower translation”. Higher density could also reflect higher initiation rates (both would lead to heavier polysomes).  In this light, what would be the cumulative effect of a series of non-GNN codons in a row?

4.        Page 7, line 226-230 – It is not clear how the preference for a GNN reduces the probability of ribosome collisions.  In large measure, this will reflect the loading of ribosomes (initiation) and subsequent elongation.  This reviewer is unaware of anything except a rare codon or an exceptionally stable RNA structure that would slow a ribosome so sufficiently as to lead to ribosome collisions.

Minor concerns

1.        The use of the word “regulates” in the title is misleading.  Regulation normally means that there is the ability to increase and decrease an event in response to some biological signal.  The GNN effect is constitutive.  This reviewer would suggest “enhances” or some other term.

2.        While the authors have several papers published using the CAR terminology, it is still useful to define this term as the abbreviation is non-standard.  By the same token, the term MD should be defined when first used also.

Reviewer 2 Report

Comments and Suggestions for Authors

In the manuscript by Sun and colleagues, the authors analyzed the distribution of codon pairs in protein-coding sequences, primarily focusing on S. cerevisiae. The results are solely based on data analysis and are not supported by biochemical validation. The authors report that the frequency distribution of two consecutive codon pairs in the ORFs deviates from what would be expected to occur purely by chance. In particular, one key observation is that the NNU codons are more likely to be immediately followed by a GNN codon. They follow up with an analysis of published ribosome profiling data and report a relative enrichment for reads with an NNU at the A site when this codon is immediately followed by a GNN. They hypothesize that this is due to interactions between the GNN and the ribosome's CAR surface, leading to slowed translocation. While the reported observations are interesting, the study has a number of limitations that call into question the interpretation favored by the authors.

Major concerns:

1. Figs. 1 and S1 are interpreted as showing an increase in G frequency in the 1st codon position, although from looking at the graphs, one might just as easily say that a G is disfavored in the second codon position. Consistent with that, in Fig. S1, A and B, the middle-G codons are noticeably the shortest bars. When looking at Fig S2 and S3, the trend of reduced middle-G appears to be universal. One might pick up other trends: for example, in eukaryotes shown in Fig. S3, an A is disfavored after NNU, but not NNC codons. Further, considering pyrimidine bases vs purines might lead to yet another set of patterns, etc. Thus, the emphasis on the GNN in this paper appears to be due to subjective preferences, and it remains unclear to what extent this specific factor determines the subtle patterns of frequency distributions observable in the data.

2. In Fig. 1, there is a fractional increase in GNN frequency after NNU in the top 10% expressers. What about the bottom 10% (or perhaps 25%)? If there is truly an effect on fitness due to enhanced translation, as proposed, one should expect the frequency to trend closer to baseline in the poorly expressed CDS's.

3. The statistical robustness of the shown differences in codon frequencies is unclear. How exactly was the bootstrapping done for the data in Fig. 1? Do the p-values refer to differences between the NNU and NNC codon sets or between the G and other bases at that position? What is the frequency of UG vs CG dinucleotides in the yeast noncoding sequences and throughout the genome? Is it statistically different from the dinucleotide frequencies observed in the NNU-GNN and NNC-GNN codon contexts? In this reviewer's opinion, these are essential metrics to demonstrate that the observed differences are not due to factors unrelated to translation.

4. In Fig. 2, ribosome footprint densities of NNUs at the A site are increased compared to NNC if they are followed by GNNs. However, it is not very clear how the calculations were performed. According to the legend, "Codon ribosome densities (RD) were normalized relative to other codons in the same mRNA." Since NNU-GNN codon pairs are shown to be more frequent in ORFs, as indicated in Fig. 1, one would expect the footprint frequencies to follow the same trend. Therefore, the authors need to provide a statistically robust method to differentiate between the presumed slowing of ribosomes (their proposed explanation) and the underlying increase in GNN frequency after NNU that they claimed earlier.

5. There could be alternative explanations for the observed phenomena that are not entertained in the discussion. For instance, one could speculate that the speed of decoding rather than translocation could be at play; in other words, the presence of G at the +1 position (or another type of interactions between the mRNA and the CAR surface) could lead to a longer dwelling time of the preceding codon. There are certainly experimental approaches that could help make a call, and their lack in this work presents a major interpretive limitation.

Minor

Lines 50-51: “This stacking of nt 34 with C1054 of CAR positions CAR as an extension of the anticodon poised to interact with the mRNA +1 codon.” - please revise this sentence.

Round 2

Reviewer 1 Report

Comments and Suggestions for Authors

The authors have responded to previous concerns and the revised manuscript is acceptable for publication.  That said, it was felt that the authors skipped over the comment about amino acids with beginning G nucleotides in their codon are "over represented" in the amino acid composition of bulk proteins.
